# A Hybrid Residential Short-Term Load Forecasting Method Using Attention Mechanism and Deep Learning

Xinhui Ji [1,2,3], Huijie Huang [1,2], Dongsheng Chen [3], Kangning Yin [3], Yi Zuo [1,2], Zhenping Chen [1,2,*] and Rui Bai [4]

1 School of Electronic and Information Engineering, Suzhou University of Science and Technology, Suzhou 215009, China
2 Suzhou Smart City Research Institute, Suzhou University of Science and Technology, Suzhou 215009, China
3 Kashi Institute of Electronics and Information Industry, Kashi 844199, China
4 State Grid Suzhou Power Supply Company, Suzhou 215004, China
* Correspondence: zhpchen@usts.edu.cn

**Abstract:** Development in economics and social society has led to rapid growth in electricity demand. Accurate residential electricity load forecasting is helpful for the transformation of residential energy consumption structure and can also curb global climate warming. This paper proposes a hybrid residential short-term load forecasting framework (DCNN-LSTM-AE-AM) based on deep learning, which combines dilated convolutional neural network (DCNN), long short-term memory network (LSTM), autoencoder (AE), and attention mechanism (AM) to improve the prediction results. First, we design a T-nearest neighbors (TNN) algorithm to preprocess the original data. Further, a DCNN is introduced to extract the long-term feature. Secondly, we combine the LSTM with the AE (LSTM-AE) to learn the sequence features hidden in the extracted features and decode them into output features. Finally, the AM is further introduced to extract and fuse the high-level stage features to achieve the prediction results. Experiments on two real-world datasets show that the proposed method is good at capturing the oscillation characteristics of low-load data and outperforms other methods.

**Keywords:** residential short-term load forecasting; deep learning; dilated convolutional neural network; long and short-term memory network; attention mechanism



## 1. Introduction

With the development of smart cities and smart homes, the daily electricity units of the residents are becoming increasingly numerous, resulting in a more complicated power system. As shown in Figure 1, a typical residential power supply includes biomass power generation, photovoltaic power generation, hydropower generation, and wind power generation. The residential electricity demand poses a huge potential threat to maintaining the stability of the power system. However, the existing energy supply structure is still dominated by thermal power generation, and the carbon emissions of thermal power generation will lead to climate warming, which, on the contrary, leads to an increase in energy demand. Hence, accurate residential power load forecasting is important. Generally, power load forecasting is divided into three categories: short-term load forecasting, medium-term load forecasting, and long-term load forecasting [1]. Short-term load forecasting can predict the sum of energy consumption within a few minutes or hours, which can optimize the energy dispatch and reduce the micro-grid primary energy loss [2]. Further, residents can reduce electricity costs by formulating electricity utilization strategies with the current pricing scheme. In addition, short-term load forecasting provides a judgment basis for some abnormal power information and guarantees the safety of people's lives and property. Therefore, in this paper, we mainly consider how to design a suitable short-term load forecasting method to improve forecasting accuracy.

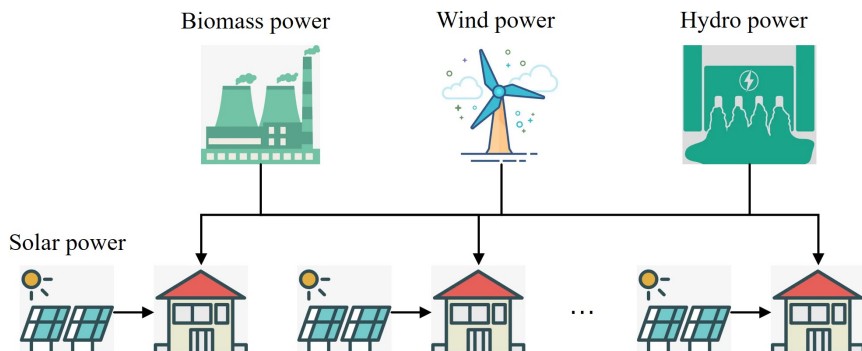

**Figure 1.** Structure of residential energy supply.

Short-term load forecasting is a time series forecasting problem. The factors that influence residential short-term load forecasting are complex and significant, including social events, electricity price adjustments, human behavior, and other uncertainties. Initially, researchers relied on artificial feature analysis to obtain empirical models, which is time-consuming and inaccurate [3]. Due to the widely different distribution of residential electricity consumption, load forecasting tends to utilize machine learning and deep learning to optimize the results. Support vector machine (SVM) [4], autoregressive integrated moving average (ARIMA) [5], extreme gradient boosting regressor (XGBoost) [6], artificial neural network (ANN) [7], and long short-term memory network (LSTM) [8] are the most commonly used methods [9]. Machine learning methods exhibit some benefits: (1) they have high computational efficiency; (2) they are highly interpretable in their basic form [10]. Although machine learning methods perform well in load forecasting, deep learning for load forecasting can obtain better results. First, deep learning does not require the creation of feature engineering in machine learning. Second, deep learning has a good generalization [11].

Currently, many studies have focused on the accuracy of short-term load forecasting. However, they have ignored trend tracking in oscillating data, especially in valley data. The oscillating data are likely to correspond to the operation of some power consumption units. If this situation cannot be predicted accurately, then for future fault monitoring and other studies, the system's stability will be challenging to achieve because of the increased likelihood of misjudgment. Therefore, this paper proposes a hybrid model, i.e., DCNN-LSTM-AE-AM, for residential short-term load forecasting. Figure 2 shows the architecture of the proposed DCNN-LSTM-AE-AM. By broadening the temporal horizon, we utilize the dilated convolutional neural network to extract temporal features from the time series. Then, the LSTM-AE is applied to mine the electricity consumption characteristics thoroughly. Finally, an attention mechanism (AM) is used to reflect the importance of behaviors in load prediction. The main contributions of this paper are listed as follows:

- Considering that individual data loss may still occur due to various conditions, we propose the T-nearest neighbors (TNN) algorithm to solve the problem of missing values, which can estimate the missing load according to the load data of adjacent similar days.
- We propose a hybrid short-term residential electricity load forecasting model (DCNN-LSTM-AE-AM). This proposed model focuses on the trend tracking of oscillating data, which can be captured with almost no delay, and provides a technical basis for predicting power failures in advance. Compared with other methods, DCNN-LSTM-AE-AM can capture the valley load data, which improves the prediction accuracy.
- The proposed DCNN-LSTM-AE-AM model is validated on two real-world datasets and compared with the existing methods. Experimental results show that this model improves the prediction results and has a good generalization.

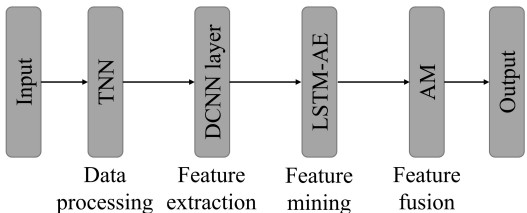

**Figure 2.** Architecture of the proposed DCNN-LSTM-AE-AM.

The rest of this paper is organized as follows. Section 2 reviews the related work. Section 3 proposes the main results on how to design a hybrid short-term residential electricity load forecasting model. In Section 4, the experimental results are demonstrated, and some comparisons with the other three models are made. Finally, Section 5 concludes the paper and future work.

## 2. Related Work

In this section, we discuss the previous research on short-term load forecasting. The existing approaches for feature extraction can be divided into three: manual feature screening, traditional machine learning, and deep earning.

In the early days, load data were less affected by uncertain factors such as human behavior and climate change. Researchers first processed the data by relying on expert experience. Then, with the application of some statistical methods, they tried to build a prediction model. For example, Taylor analyzed the seasonal cycle within the day and week as the key features and processed them with five statistical methods to obtain an optimal forecasting result [12]. Statistical methods are rich in theory and highly interpretable for some specific characteristics. However, due to the growth in the living standard, the approaches are difficult for experts to interpret the complex data.

With the advance of big data and artificial intelligence, data-driven load forecasting technologies have received attention extensively [13–17]. The computing capacity of hardware devices has greatly influenced short-term load forecasting. He et al. [18] utilized ARIMA to design a high-frequency short-term load forecasting model. The model divided the inputs by season and then used hourly load data to predict energy consumption for the next month. Cao et al. [19] divided the dataset by the characteristics of similar daily meteorological conditions and applied ARIMA for prediction. Although the above models predicted well in the same season, such models based on seasons are extremely dependent on manual screening and do not have the ability to generalize. Cai et al. [20] proposed an energy prediction model that combined the K-means and data mining methods to analyze the energy consumption of 16,000 residential buildings. Mohammadi et al. [21] proposed a hybrid model based on sliding window empirical mode decomposition (SWEMD) to predict the power consumption of small buildings. They also proposed an algorithm to optimize the model parameters. Chauhan et al. [22] designed a hybrid model based on SVM and ensemble learning for prediction, where load data were processed separately through hourly and daily resolutions. Experiments in Aguilar Madrid and Antonio [6] showed that the XGBoost could obtain the best performance from the machine learning algorithms set. Massaoudi et al. [23] proposed an ensemble-based LGBM-XGB-MLP hybrid model to improve prediction performance. Although traditional machine learning models can achieve good results in load forecasting, they all require more effort in the feature selection and parameter optimization process [11].

Deep learning has received explosive growth in various fields. Some typical networks for deep learning, such as ANN [24–26], convolutional neural network (CNN) [27], recurrent neural network (RNN) [28], and LSTM [29], have been widely used in load forecasting. Chen et al. [30] added periodic features and utilized a deep residual network (ResNet) to predict the hourly residential load, whose evaluation performance had been improved. From the aspect of demand-side management (DSM), Kong et al. [31] proposed an improved deep belief network (DBN) method to forecast 1-h-level loads. Compared with

others, this method could significantly improve both the day-ahead and week-ahead load forecasting results. Dong et al. [32] proposed a distributed deep belief network (DDBN) with Markov switching topology, which improved distributed communication stability and prediction accuracy.

It should be noted that energy consumption is generated by a variety of complex factors, so how to utilize the nonlinear models to improve prediction accuracy is the main consideration. Recently, CNN has injected vitality into short-term load forecasting [33–35]. Amarasinghe et al. [36] tried to apply CNN for prediction on the residential load dataset [37]. Experimental results show that this method is effective, but the prediction accuracy needs to be improved. Sadaei et al. [38] proposed a hybrid algorithm combining CNN and fuzzy time series (FTS) for forecasting. This method converted multivariate time series into multi-channel images. The proposed model overcame some advanced time series models for short-term load forecasting with better results and solved the problem of over-fitting.

RNN, LSTM, and gated recurrent neural network (GRU) are designed for time-series data, which is sensitive to temporal features. They can fully mine time-related features between adjacent data. Rahman et al. [39] utilized RNN to forecast the 1-h commercial and residential load data in the medium and long-term forecasting and thus realized the load trend tracking. However, it is challenging for RNNs to achieve convergence on data within a long time interval. Kong et al. [8] relied on the multi-layer LSTM to predict individual energy consumption. Compared with others, Kong et al. [8] not only solved the problem of RNN but also improved the prediction accuracy. Li et al. [40] proposed an improved GRU to dynamically capture temporal correlations within the forecast period, enabling the model to adapt to different datasets.

In the field of load forecasting, it is more inclined to utilize the feature extraction capabilities of the hybrid model to improve forecasting accuracy. Shi et al. [41] proposed a pooling method based on the LSTM network, which further improved the prediction accuracy of LSTM through backpropagation and pooling. Jiang et al. [42] proposed a hybrid model based on CNN and LSTM to predict household energy consumption. Unlike other combined methods, this method divided the input into two parts: long and short data. Then, they obtained a result with sufficient feature interaction through data fusion. Yue et al. [43] combined ensemble empirical mode decomposition (EEMD), permutation entropy (PE), feature selection (FS), LSTM, and bayesian optimization algorithm (BOA) to optimize the prediction accuracy. Further, some reasonable explanations were made for the reconstructed subsequences. Lin et al. [44] proposed an AM-based auto-encoder structure for LSTM, which provided superior prediction accuracy and had a good generalization. Wei et al. [45] proposed detrend singular spectrum fluctuation analysis (DSSFA) to extract trend and periodic components and then input these components into LSTM to improve short-term forecasting accuracy. Laouafi et al. [46] proposed an adaptive hybrid ensemble method named CMKP-EG-SVR and optimized the result of the mixture model through a gaussian-based error correction strategy.

## 3. Model Architecture of Residential Short-Term Load Forecasting

The goal of residential short-term load forecasting is to improve residents' electricity experience. It is an essential part of energy supply management. In this section, we mainly introduce a complete forecasting architecture that can improve residential load forecasting accuracy.

Figure 3 shows the process of residential short-term load forecasting, DCNN-LSTM-AE-AM. In Figure 3, the missing data are first processed based on the TNN algorithm, and the DCNN layer extracts the initial features. Then, the LSTM-AE is used to extract the spatiotemporal feature information. Finally, the AM is introduced to analyze the importance of the extracted features and outputs the final prediction result.

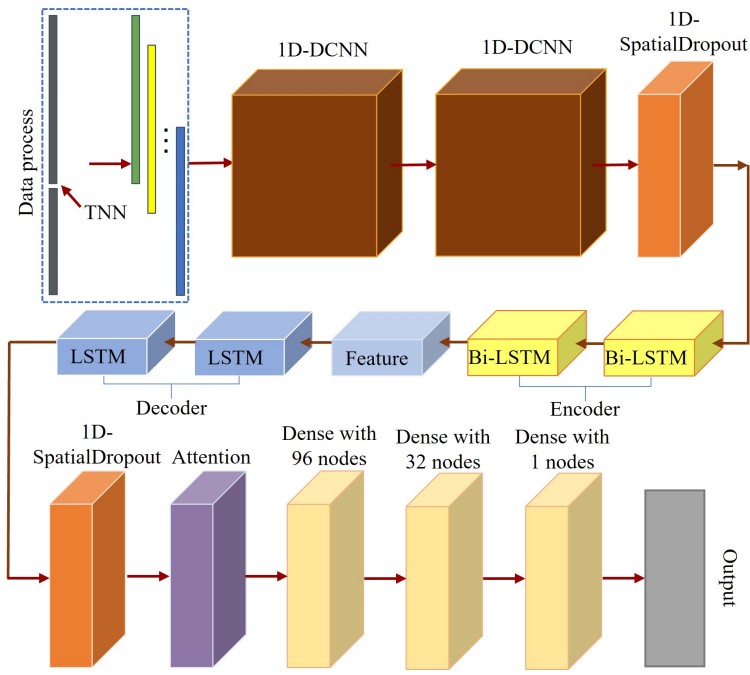

**Figure 3.** Process of residential short-term load forecasting.

### 3.1. Data Processing

Residential energy consumption data are collected by smart meters. Smart meters can accurately collect and transmit data to the data management center through various communication networks. Considering that the communication network is susceptible to interference by multiple factors, data loss is inevitable. Therefore, the missing data needs to be processed by some methods. This paper proposes the T-nearest neighbors (TNN) algorithm to fill in the missing data. At time $t$, TNN is defined as follows:

$$I_t \leftarrow \frac{1}{K}\left(I_{t-\frac{K}{2}T} + I_{t-(\frac{K}{2}-1)T} + \cdots + I_{t-T} + I_{t+T} + \cdots + I_{t+(\frac{K}{2}-1)T} + I_{t+\frac{K}{2}T'}\right) \quad (1)$$

where $K$ represents the number of selected adjacent values; $T$ represents the interval period; $I_t$ is the output of TNN at time $t$. The algorithm can solve the problem that the duration of missing data is relatively long. When the duration of the missing data is too long, the missing data will be ignored.

Moreover, the collected load data usually has a small amount of singular data, which will affect the overall model. Therefore, it is necessary to scale these data to some fixed range so that they will conform to a certain distribution. In this paper, we use a linear normalization, i.e., the max-min normalization, to process the load data, which is defined as follows:

$$\theta_{norm} = \frac{\theta - \theta_{\min}}{\theta_{\max} - \theta_{\min}}, \quad (2)$$

where $\theta_{norm}$ represents the normalized output; $\theta$ represents the current input; $\theta_{\max}$ and $\theta_{\min}$ represent the upper and lower bounds of the current sequence input, respectively.

### 3.2. Dilated Convolutional Neural Network

CNN has been widely used in image processing ever since it was proposed [36]. Recently, time series data has also tried to use CNN to deal with short-term load forecasting. The core of CNN is weight sharing. Each CNN has a convolution kernel, which shares different weights according to the convolution operation. However, CNN will lose this feature information with long-term regularity, such as valley oscillation load [47]. To broaden the horizon of CNN, in this paper, we transform the convolution computation of continuous data into the convolution computation of skipping data, which is called the

dilated convolutional neural network (DCNN) [48,49]. For a $\tau$-dimension input vector $\nu \in \mathbb{R}^\tau$ and a kernel $w : \{0, \ldots, k-1\} \in \mathbb{R}$, the dilated convolution operation on element $s$ of input vector $\nu$ is defined as:

$$y_s = \sum_{i=0}^{k-1} \nu[s + r \cdot i] \cdot w[i], \tag{3}$$

where the dilated convolution adjustment rate $r$ is expressed as the interval step size for selecting input data; $k$ represents the kernel size. In addition, $\tau$ represents the time step.

Figure 4a,b illustrate the internal structures of CNN and DCNN, respectively. In Figure 4, CNN has one dimension, a kernel of one, and a dilation rate of one, while DCNN has one dimension, a kernel of one, and a dilation rate of one. The first layer is the original input layer, the second layer is the hidden layer, and the third layer is the output layer. In addition, after adding the convolution layer, the activation function and pooling layer are added appropriately to help the backpropagation of the gradient.

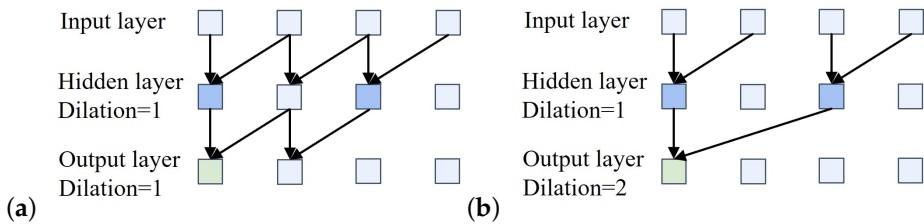

**Figure 4.** Structure of CNN and DCNN: (**a**) CNN; (**b**) DCNN.

### 3.3. LSTM-Based Autoencoder

#### 3.3.1. Long Short-Term Memory Network

RNN is a forward-propagating sequential neural network. When it deals with long-duration data, it usually faces some challenges, such as gradient disappearance and gradient explosion. Hochreiter and Schmidhuber [29] proposed an improved network based on the RNN structure and named it LSTM. The internal structure of the LSTM is shown in Figure 5, where the memory cell can retain information from a long time ago, and the forget gate can choose to discard some feature information. Backpropagation in the LSTM strengthens the interaction ability of context information and reserves more useful spatiotemporal feature information.

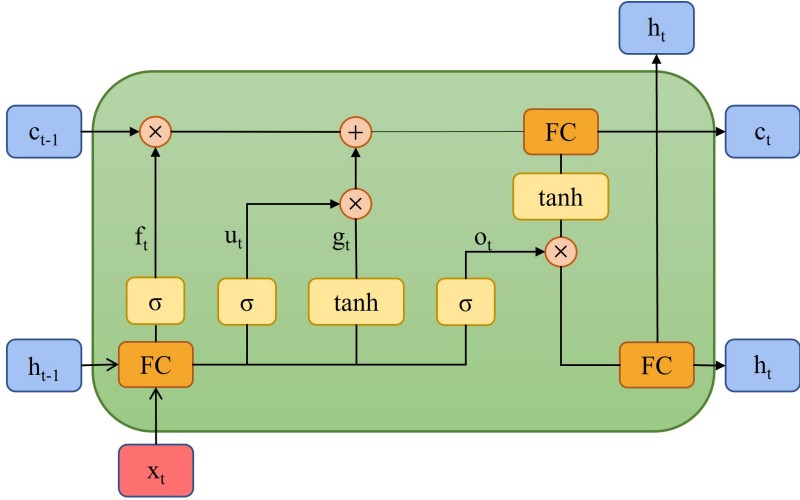

**Figure 5.** Internal structure of LSTM.

The update principles of the LSTM are defined as:

$$f_t = \sigma\left(W_{ft}x_t + W_{fh}h_{t-1} + b_f\right), \tag{4}$$

$$u_t = \sigma(W_{tx}x_t + W_{uh}h_{t-1} + b_u), \tag{5}$$

$$g_t = \tanh\left(W_{gx}x_t + W_{gh}h_{t-1} + b_g\right), \tag{6}$$

$$o_t = \sigma(W_{ox}x_t + W_{oh}h_{t-1} + b_o), \tag{7}$$

$$c_t = g_t \odot u_t + c_{t-1} \odot f_t, \tag{8}$$

$$h_t = \tanh(c_t) \odot o_t, \tag{9}$$

where $f_t$, $u_t$, $g_t$, $o_t$ in Equations (4)–(7) represent the information at the forget gate, input gate, input node, and output gate at time $t$; $\sigma$, tanh are the multiplication calculations and activation functions; $b_f, b_u, b_g, b_o$ are the bias parameters of the corresponding processing units; $c_t, h_t$ in Equations (8) and (9) represent memory cells; $W_{ft}$, $W_{fh}$, $W_{tx}$, $W_{uh}$, $W_{gx}$, $W_{gh}$, $W_{ox}$, and $W_{oh}$ are the weight matrices of the corresponding processing units; $\odot$ represents element multiplication; FC represents the fully connected layer. These units use functions $\sigma$ and *tanh* to continuously compress the input $x_t$ to a smaller range.

### 3.3.2. Bidirectional Long Short-Term Memory Network

The bidirectional long short-term memory network (BiLSTM) is a variant of LSTM. As shown in Figure 6, BiLSTM is a special network structure formed by superimposing two LSTM layers. The LSTMs synchronously train the input data at the same time step. These two LSTM layers differ in that one input the data in a positive temporal order, and the other processes it in a reverse temporal order. This structure not only utilizes the information of the previous moment but also relies on the information of the latter moment [50,51].

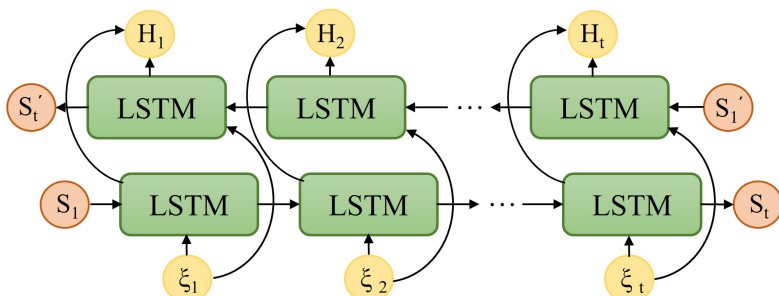

**Figure 6.** Standard structure of BiLSTM.

Let $\overrightarrow{h_t}$ and $\overleftarrow{h_t}$ be the hidden states of forward and backward propagation, respectively. Then, $\overrightarrow{h_t}, \overleftarrow{h_t}$, and the output $H_t$ of BiLSTM are calculated as follows:

$$\overrightarrow{h_t} = \overrightarrow{LSTM}(\xi_t, S_{t-1}), t \in [1, \mathbb{T}], \tag{10}$$

$$\overleftarrow{h_t} = \overleftarrow{LSTM}(\xi_t, S_{t+1}), t \in [\mathbb{T}, 1], \tag{11}$$

$$H_t = FC\left(\overrightarrow{h_t}, \overleftarrow{h_t}\right). \tag{12}$$

where $\xi_t$ represents the current input at current time $t$; $S_t$ represents the internal state in the LSTM, that is, the memory cells and the hidden state; $\mathbb{T}$ represents the time steps.

### 3.3.3. Autoencoder

Autoencoder (AE), as an artificial neural network (ANN), is a conceptual network structure with an encoder and decoder. The AE aims to find an optimal set of connection weights by minimizing the reconstruction error between the original input and the out-

put [52]. For any AE, there is a $n$-dimensional input vector $\varphi_t$ and an output vector $\epsilon_t$ with random dimensions. The input $\varphi_t$ can be mapped to output $\epsilon_t$ according to the following mapping functions:

$$\Theta = f(\tilde{W} \cdot \varphi_t + \alpha), \tag{13}$$

$$\epsilon_t = \rho(\hat{W} \cdot \Theta + \beta), \tag{14}$$

where $\Theta$ is the mapping output of the encoding layer; $\tilde{W}$ and $\hat{W}$ are two weight matrices; $f$ and $\rho$ are two activation functions; $\alpha$ and $\beta$ are the bias parameters of the encoder and decoder, respectively.

Obviously, simple nonlinear AE is difficult for time series data feature extraction. As shown in Figure 3, in this paper, the LSTM and BiLSTM are combined to construct the AE. In order to learn more feature information and increase the sensitivity of the contextual information connection, the encoder part adopts two BiLSTM layers. In addition, the increase in the depth of the neural network structure is conducive to the extraction and fusion of load features. In the decoder part, we only need to add two LSTM layers as the feature analysis layer to reduce the unnecessary network computing burden.

### 3.3.4. Attention Mechanism

The AM pays more attention to the important parts of reconstructing the cognitive world instead of making an average judgment on the whole. Figure 7 shows the structure of the AM. In Figure 7, FC is the fully connected layer, which fuses with the output of the AM. The output of the AM can be calculated as follows:

$$\mu_t = \sum \Omega_t \odot \eta_t. \tag{15}$$

where $\eta_t = \{h_1, \cdots, h_\chi\}$ is the output decoded by LSTM-AE and is a $\chi$-dimensional hidden state vector at time $t$; $\Omega_t = \{\lambda_1, \cdots, \lambda_\chi\}$ is a weight matrix.

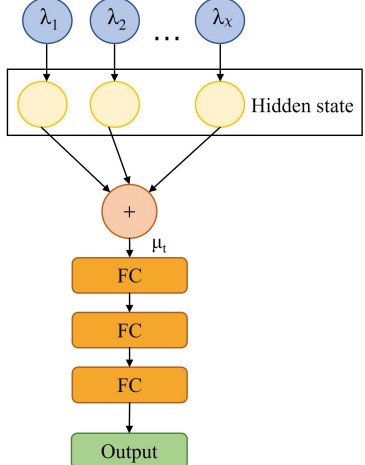

**Figure 7.** Structure of attention mechanism.

The matrix $\Omega_t$ in (15) can be implemented according to the following procedure. First, the output $\eta_t$ represents the input of the AM. Then, the alignment model $a(\cdot)$ aligns the input with the output vector $\phi_t = \{\varepsilon_1, \cdots, \varepsilon_\chi\}$. The alignment score $\phi_t$ is calculated as follows:

$$\phi_t = a(\delta_{t-1}, \eta_t). \tag{16}$$

In this study, the alignment model $a(\delta_{t-1}, \eta_t)$ represents $\tanh(\delta_{t-1} \odot \eta_t + \gamma)$, where the cell state $\delta_{t-1}$ decoded by LSTM-AE represents the $\chi$-dimensional hidden state vectors

at time $t - 1$; $\gamma$ is a vector of bias parameters. Finally, each element $\lambda_j$ is computed by applying a softmax operation:

$$\lambda_j = \frac{\exp(\varepsilon_j)}{\sum_{i=1}^{\chi} \exp(\varepsilon_i)}, \tag{17}$$

where $i$ and $j$ represent the $i$-th and $j$-th elements in $\phi_t$. After injecting the AM, we also apply multiple fully connected layers as the output layer to complete the final load forecasting.

## 4. Experiment Results and Analysis

In this section, we will show the effectiveness of our proposed DCNN-LSTM-AE-AM load forecasting method through some experiments.

### 4.1. Experiment Settings

4.1.1. Dataset Selection

The experiments use two real-world power load datasets to test our proposed method's robustness and generalization. The household electricity consumption dataset from the UCI machine learning library (IHEPC) [37] records the energy consumption information of a house from 2006 to 2010. It contains multiple attributes: date, time, global active power, global reactive power, voltage, current, and active power of three-room types: kitchen, bathroom, and bedroom. There are 2,075,269 1-min-level data, including 25,979 missing values. Table 1 shows detailed information on IHEPC. The global active power represents the actual power consumption, so this paper only takes the global active power as the input. Another dataset is from the smart grid and smart city (SGSC) [53] projects carried out by the Australian government and industry consortium Ausgrid. Data in the SGSC dataset are collected from 10,000 households and some retail stores in New South Wales (NSW) from 2010 to 2014. In this paper, the IHPEC is organized as the sum of energy consumption within 15 min and 1 h. Since the SGSC only provides electricity consumption per half hour, the SGSC is organized as the sum of energy consumption within 30 min and 1 h. Figure 8 shows the randomly selected data of IHPEC at a 15-min resolution and the randomly selected data of one household from SGSC at a 30-min resolution.

**Table 1.** Information from the IHEPC dataset.

| Variable | Description |
| --- | --- |
| Data | Recorded date |
| Time | Current moment |
| Global active power | Sum of active power per minute |
| Global reactive power | Sum of reactive power per minute |
| Voltage | Voltage per minute |
| Global intensity | Sum of current per minute |
| Sub metering1 | Power used by kitchen per minute |
| Sub metering2 | Power used by room per minute |
| Sub metering3 | Power used by bathroom per minute |

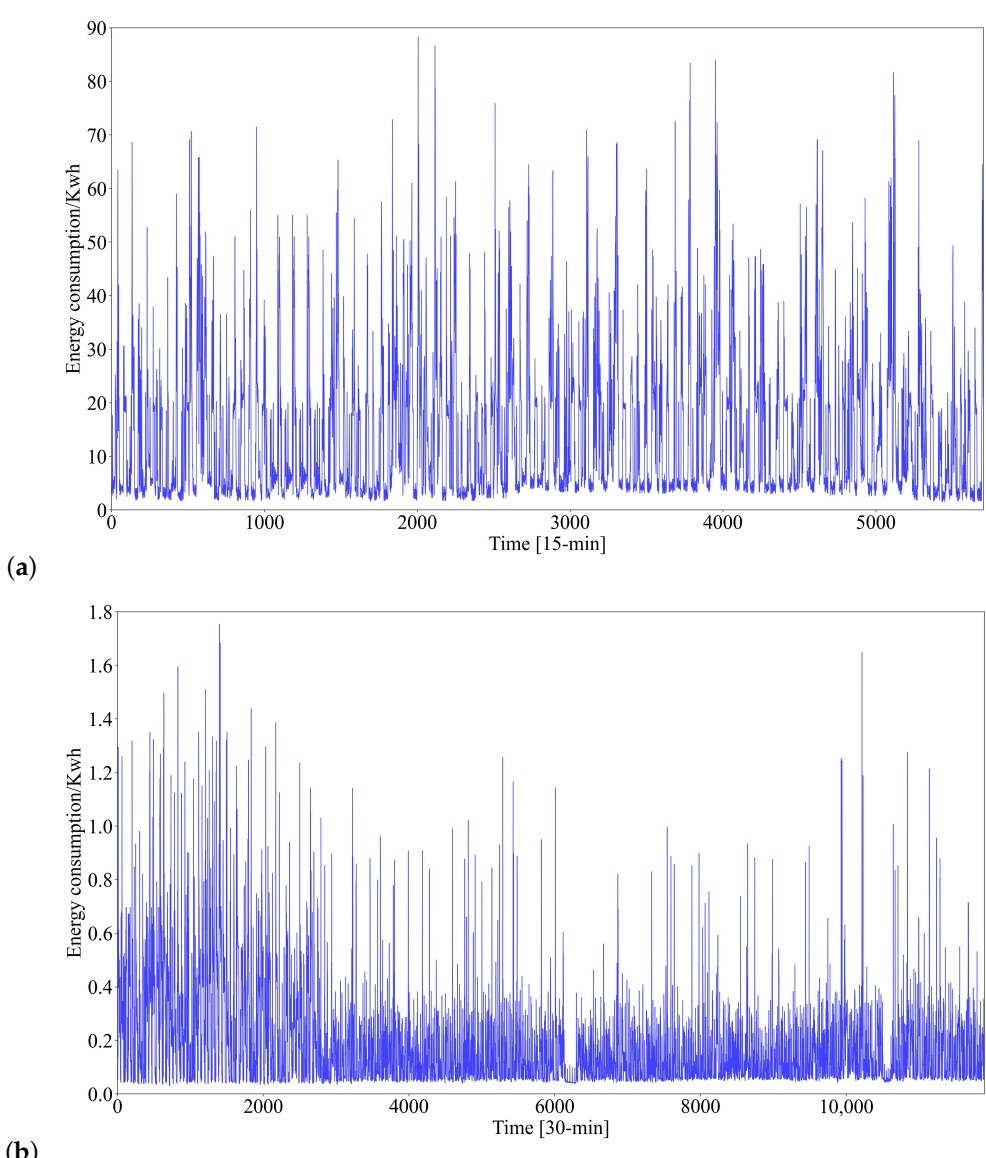

**Figure 8.** Visualization of selected datasets: (**a**) IHEPC at a 15-min resolution; (**b**) SGSC at a 30-min resolution.

### 4.1.2. Experiment Setup

All networks are built based on Python3.6, Keras2.2, and TensorFlow2.0. The device is configured with a 2.6 GHz intel i9 CPU and a 16GB NVIDIA TESLA T4 GPU.

According to the findings in [8], some rules of thumb for hyperparameter selection are adopted. Since hyperparameter selection is a time-consuming task, in this paper, we try to use different combinations of parameters to obtain the optimal performance in MSE. Table 2 lists the hyperparameter settings of the proposed DCNN-LSTM-AE-AM. The first 1D-Conv layer uses the DCNN with a kernel of 3, 12 filters, and a dilation rate of 2 to extract features and ReLU as output results. The number of the second 1D-Conv layer's filters is upgraded to 24. These two SpatialDropout layers randomly zero the parameters with probabilities 0.1 and 0.2, respectively. In the LSTM-AE, 32 units are used in all four temporal models. In the final output layer, the AM uses 32 units, and the three Dense layers use 96, 32, and 1 unit, respectively. We set the maximum training epoch to 50. All other methods are also tested on the same equipment, hyperparameter configuration, and environment to allow horizontal comparisons. In addition, XGBoost obtains optimal performance with the number of estimators set to 30 after constantly changing the number of estimators.

In this paper, we split the dataset into a training set and a test set, whose ratios are designed as 0.67 and 0.33. We fill the missing values in the training set according to the TNN algorithm. The missing values in the test set are ignored to prevent prior knowledge leakage.

**Table 2.** Hyperparameter settings.

| Network | Hyperparameters |
| --- | --- |
| 1D-Conv | The convolution kernel size is 3, the number of filters is 12, the dilation rate is 2, and the activation function is ReLU |
| 1D-Conv | The convolution kernel size is 3, the number of filters is 24, the dilation rate is 2, and the activation function is ReLU |
| 1D-SpatialDropout | 0.1 |
| BiLSTM | 32 units |
| BiLSTM | 32 units |
| LSTM | 32 units |
| LSTM | 32 units |
| 1D-SpatialDropout | 0.2 |
| Attention | 32 units |
| Dense | 96 units |
| Dense | 32 units |
| Dense | 1 unit |

### 4.1.3. Evaluation Metric

It is well known that the classification task can be evaluated by accuracy in percentage. However, this kind of accuracy is not appropriate for evaluating any regression task. In this paper, in order to objectively evaluate the fairness and integrity of the methods, we use four evaluation metrics: MAE, RMSE, MSE, and MAPE. The MAE is a method of averaging quantization errors. Compared with MAE, the other methods have made some improvements. MSE pays more attention to the influence of outliers on the overall prediction effect. RMSE performs arithmetic square root on the overall basis of the MSE, which amplifies the difference of the MSE. MAPE can focus on the gap between the error and the actual value. The formulations for the MAE, RMSE, MSE, and MAPE are listed as follows:

$$MAE = \frac{1}{N} \sum_{i=1}^{N} \left| \psi_{pred} - \psi \right|, \tag{18}$$

$$MSE = \frac{1}{N} \sum_{i=1}^{N} (\psi_{pred} - \psi)^2, \tag{19}$$

$$RMSE = \sqrt{\frac{1}{N} \sum_{i=1}^{N} (\psi_{pred} - \psi)^2}, \tag{20}$$

$$MAPE = \frac{100\%}{N} \sum_{i=1}^{N} \left| \frac{\psi_{pred} - \psi}{\psi} \right|. \tag{21}$$

where $N$ represents the total amount of the load data; $\psi$ and $\psi_{pred}$ represent the actual load and the predicted load, respectively. For all those evaluation metrics, we have that the smaller the value, the more accurate the model.

### 4.2. Influence of Hyperparameters

This subsection sets various optimized hyperparameters to achieve the best performance. We evaluate the performance of the proposed model under these hyperparameters by the MSE. To ensure the fairness of the experiments, we compare the influence of batch size, optimizer, and learning rate on the MSE with other parameters that remain unchanged. Figure 9 shows the MSE of the proposed method under different hyperparameter settings. From Figure 9, we can have that our method has a certain degree of sensitivity to hyperpa-

rameters. According to the experimental results in Figure 8, we choose the batch size of 64, the optimizer as Adam, and the learning rate as 0.001.

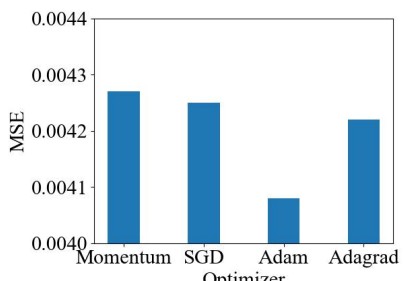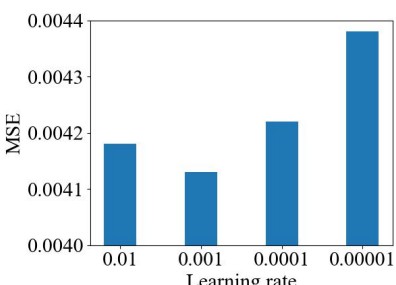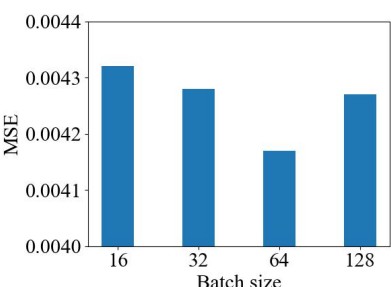

**Figure 9.** MSE of the proposed model under different hyperparameters.

### *4.3. Influence of Time Step*

Time step $\tau$, i.e., the length of the input data, is one of the factors affecting the robustness and accuracy of the proposed short-term load forecasting model. In deep networks, long time-series data often results in overfitting. Hence, the data length also affects the performance of our proposed method. The MSE, RMSE, MAE, and MAPE values under different data lengths are shown in Table 3. The evaluation metrics with lengths between 8 and 14 have very little difference. Therefore, in these following experiments, we let the data length be 12.

**Table 3.** Performance of our proposed method under different data lengths.

| Length | MSE | RMSE | MAE | MAPE |
|--------|---------|--------|--------|--------|
| 6 | 0.00434 | 0.0659 | 0.0351 | 0.7599 |
| 8 | 0.00420 | 0.0648 | 0.0332 | 0.8521 |
| 10 | 0.00418 | 0.0647 | 0.0344 | 0.7662 |
| 12 | 0.00411 | 0.0641 | 0.0331 | 0.6175 |
| 14 | 0.00415 | 0.0644 | 0.0334 | 0.6757 |
| 16 | 0.00426 | 0.0653 | 0.0338 | 0.7421 |
| 18 | 0.00444 | 0.0666 | 0.0348 | 0.7952 |
| 20 | 0.00450 | 0.0671 | 0.0357 | 0.7853 |
| 22 | 0.00458 | 0.0677 | 0.0366 | 0.8322 |
| 24 | 0.00446 | 0.0668 | 0.0370 | 0.8964 |

### *4.4. Performance Evaluation on IHEPC Dataset*

The performance of the proposed DCNN-LSTM-AE-AM is compared with some sole models and some hybrid models. The sole models include XGBoost, DCNN, LSTM [8], and AM; while the hybrid models include LSTM-AE [54], CNN-LSTM [42], DCNN-AM, DCNN-LSTM-AE, and LSTM-AE-AM. The evaluation metrics comparison among different methods at 15-min and 1-h resolutions are shown in Table 4. From Table 4, we have that: (1) our proposed method obtains the best metrics of 0.0041 in MSE, 0.0640 in RMSE, 0.0333 in MAE, and 0.6757 in MAPE at the 15-min resolution; (2) while for the dataset at the 1-h resolution, our proposed method outperforms others with metrics of 0.0086 in MSE, 0.0926 in RMSE, 0.0667 in MAE, and 0.7257 in MAPE at the 1-h resolution. (3) Compared with the existing methods, the overall prediction accuracy of the DCNN-LSTM-AE-AM at different resolutions has obtained a little improvement. (4) Performances of those sole models are worse than those of hybrid models. It is worth noting that there is not much difference between the performance of DCNN-AM, LSTM-AE-AM, and DCNN-LSTM-AE-AM.

**Table 4.** Prediction performance comparisons on the SGSC dataset.

| Method | Resolution | MSE | RMSE | MAE | MAPE |
|---|---|---|---|---|---|
| XGBoost | 15 min | 0.0463 | 0.2152 | 0.1541 | 0.8683 |
| | 1 h | 0.0410 | 0.2025 | 0.1120 | 0.7268 |
| DCNN | 15 min | 0.0403 | 0.2007 | 0.1192 | 0.6869 |
| | 1 h | 0.0412 | 0.2030 | 0.1132 | 0.7284 |
| LSTM | 15 min | 0.0491 | 0.2216 | 0.1263 | 1.0384 |
| | 1 h | 0.0548 | 0.2341 | 0.1356 | 1.4831 |
| AM | 15 min | 0.0663 | 0.2575 | 0.1228 | 1.4286 |
| | 1 h | 0.0687 | 0.2621 | 0.1346 | 1.3788 |
| LSTM-AE | 15 min | 0.0179 | 0.1338 | 0.0855 | 0.7863 |
| | 1 h | 0.0232 | 0.1523 | 0.0878 | 0.8265 |
| CNN-LSTM | 15 min | 0.0158 | 0.1257 | 0.0712 | 0.6930 |
| | 1 h | 0.0197 | 0.1403 | 0.0997 | 0.7556 |
| DCNN-AM | 15 min | 0.0081 | 0.0900 | 0.0452 | 0.6938 |
| | 1 h | 0.0091 | 0.0954 | 0.0466 | 0.7028 |
| DCNN-LSTM-AE | 15 min | 0.0222 | 0.1490 | 0.0709 | 0.7380 |
| | 1 h | 0.0295 | 0.1718 | 0.0823 | 0.7428 |
| LSTM-AE-AM | 15 min | 0.0043 | 0.0656 | 0.0378 | 0.6854 |
| | 1 h | 0.0095 | 0.0975 | 0.0682 | 0.7530 |
| DCNN-LSTM-AE-AM | 15 min | 0.0041 | 0.0640 | 0.0333 | 0.6757 |
| | 1 h | 0.0086 | 0.0927 | 0.0667 | 0.7257 |

To show the effect of individual data in the model, we also use four box-figures to show the performance of different models in MSE, RMSE, MAE, and MAPE. As shown in Figure 10, four subfigures are used to show the performance of the IHEPC dataset at a 15-min resolution. Since we have given the average results in Table 4, the average results are no longer marked in the subfigures, and only the median results are marked. From Figure 10, one can have that: (1) the proposed method outperforms others in all the evaluation metrics; (2) the median of the metrics is lower than that of other methods; (3) the MSE, RMSE, and MAE obtain significant improvement, while the increase in MAPE is relatively small.

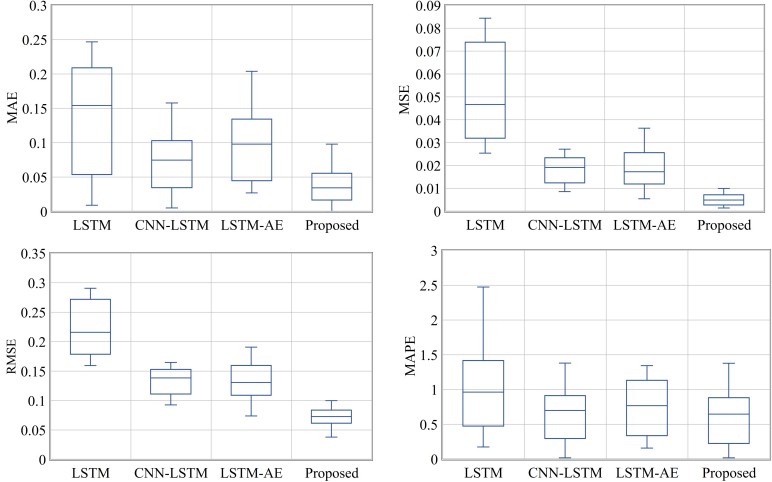

**Figure 10.** Prediction results of different methods on the IHEPC dataset at a 15-min resolution.

Figure 10 also shows that all the mentioned forecasting methods are still far from accurate prediction, which is the direction we need to focus on. Further, CNN-LSTM, LSTM-AE, and our proposed method all inherit the feature extraction capability of LSTM. To demonstrate our proposed model's trend-tracking capability, we show the prediction results of the mentioned four sole models and the six hybrid models in Figures 11 and 12. From Figure 11, it can be seen that the tracking trend of deep learning methods, except

for the AM, can outperform traditional machine learning. DCNN is always trying to match the low-load data. The LSTM has outstanding temporal feature extraction and good trend tracking, but the error with respect to the actual load is large. Although the overall performance of using only AM is not good, the AM is suitable for forecasting low-load data. The performance of AM benefits from local feature fusion capability.

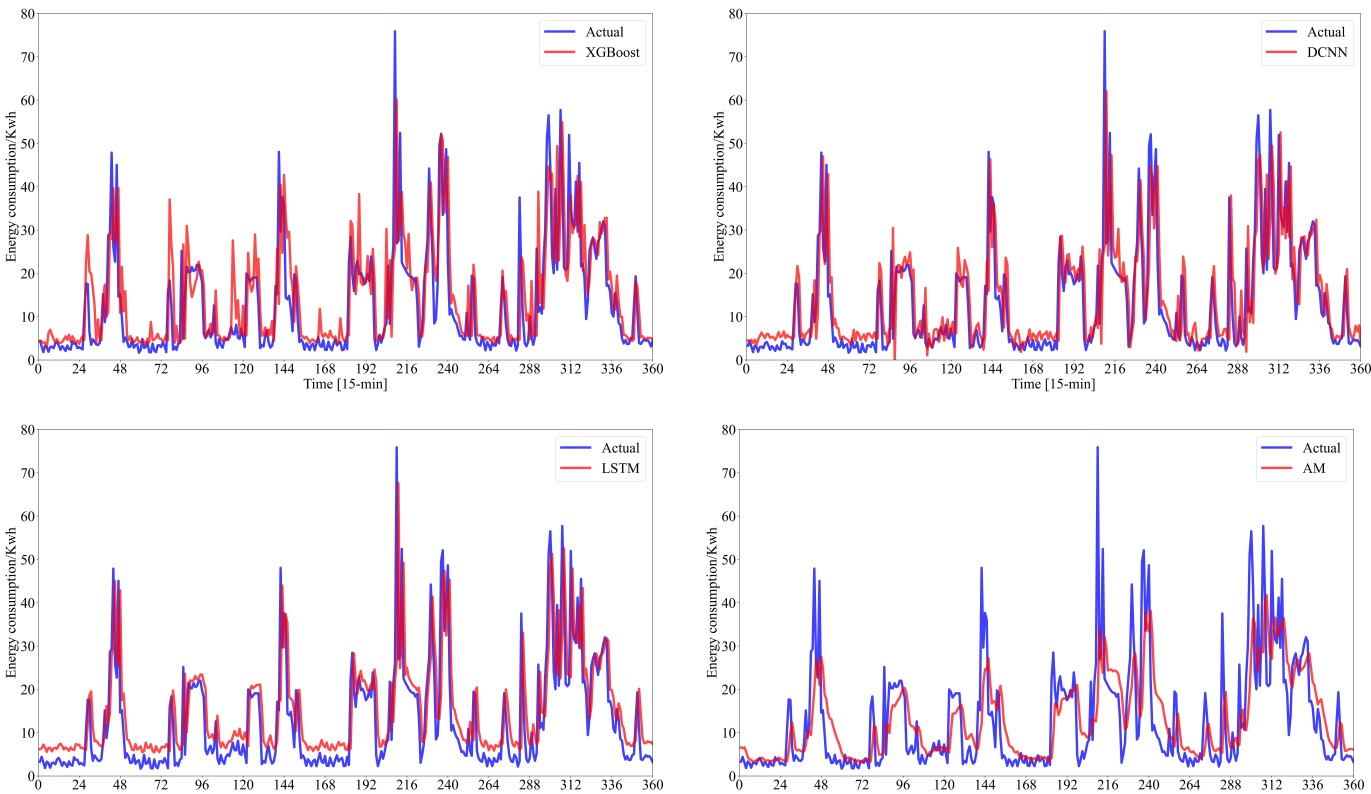

**Figure 11.** Prediction results of some sole models on the IHEPC dataset at a 15-min resolution.

From Figure 12, one can have that: (1) Different hybrid models have different prediction results. (2) The LSTM-AE is similar to the LSTM, which further narrows the numerical gap. (3) Although the CNN-LSTM is sensitive to data with large fluctuations in value and predicts the result with a small error, there is a large error when capturing the valley values. This is one of the reasons for the large value of MAPE. (4) DCNN-AM and LSTM-AE-AM are increasingly perceptive to the low-load data. The addition of AM in the last layer is definitely the main contributor to improving the prediction performance of the hybrid model on the low-load data. (5) Compared with the other five mentioned hybrid models, DCNN-LSTM-AE-AM can quickly capture the data in this situation and predict the valley data very well. This benefits from the broadened time horizon of the DCNN. The introduction of AM makes the weights closer to the actual load, which significantly reduces the prediction error. In addition, the results of DCNN-LSTM-AE suggest that no combination of hybrid models can improve the prediction results.

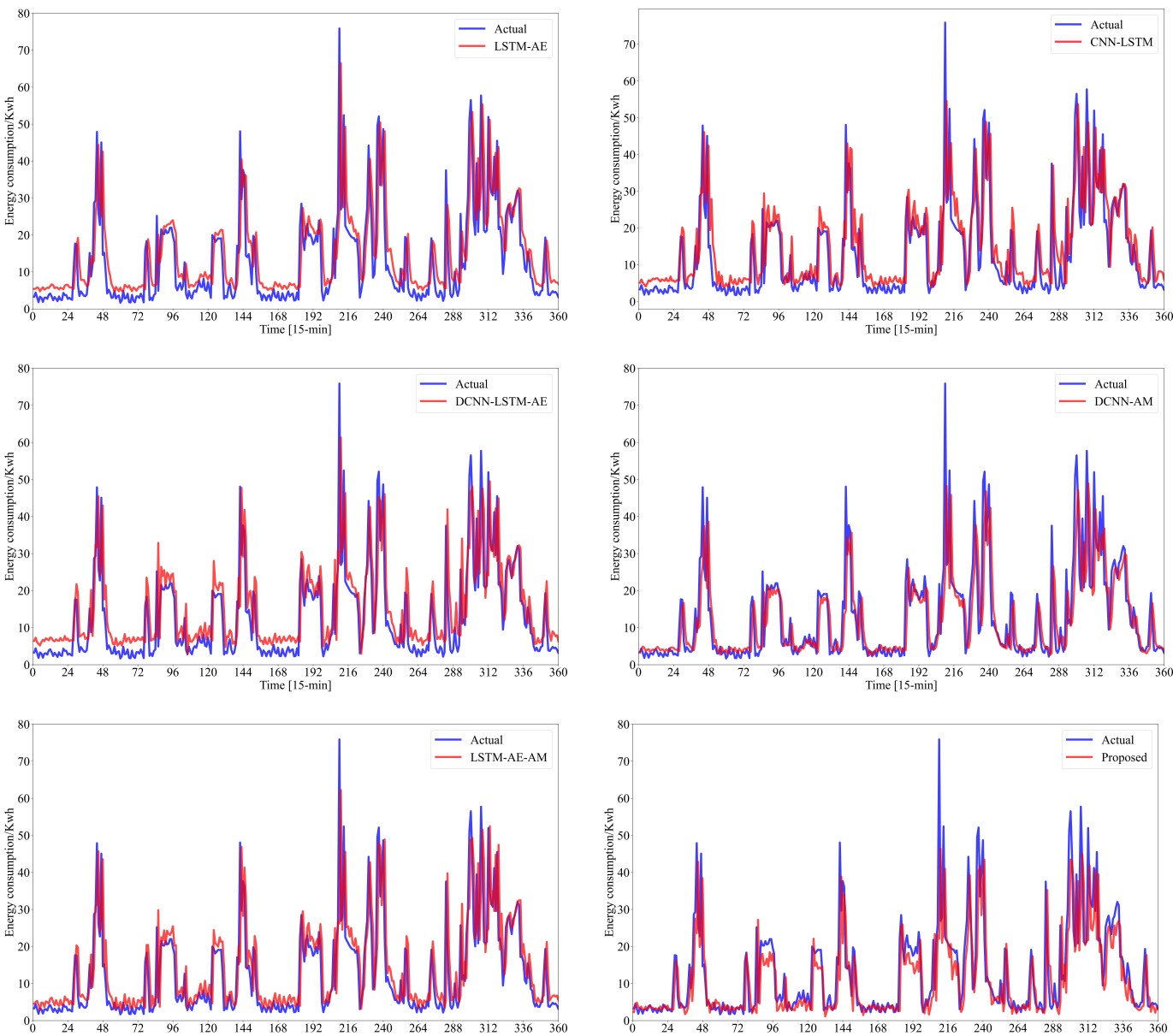

**Figure 12.** Prediction results of some hybrid models on the IHEPC dataset at a 15-min resolution.

### 4.5. Performance Evaluation on SGSC Dataset

Our proposed method is also tested on the SGSC dataset, where it shows excellent accuracy and has a good generalization. Table 5 shows the evaluation metrics comparisons of different forecasting methods on the SGSC dataset. The performance of the proposed method on the SGSC dataset is similar to that on the IHEPC dataset, and both have obtained the best results for each evaluation metric. The MSE of the proposed method obviously gets improved compared with other methods. The other evaluation metrics also have a small increase.

As shown in Figure 13, we compare the actual load with the prediction results. In this study, a residential user is arbitrarily selected, and the proposed DCNN-LSTM-AE-AM can easily predict the load at the next moment. The proposed method can capture the overall trend of the actual load with small time offsets and numerical errors. In addition, the method also has good predictability for the valley data. This confirms that DCNN and AM have a good ability to capture long-term regular data.

**Table 5.** Prediction performance comparisons on the SGSC dataset.

| Method | Resolution | MSE | RMSE | MAE | MAPE |
|---|---|---|---|---|---|
| XGBoost | 30 min | 0.0456 | 0.2135 | 0.1203 | 0.9298 |
| | 1 h | 0.0403 | 0.2007 | 0.0980 | 0.8296 |
| DCNN | 30 min | 0.0433 | 0.2081 | 0.1329 | 0.6796 |
| | 1 h | 0.0465 | 0.2156 | 0.1366 | 0.7862 |
| LSTM | 30 min | 0.0483 | 0.2198 | 0.1298 | 1.1001 |
| | 1 h | 0.0522 | 0.2285 | 0.1401 | 1.5131 |
| AM | 30 min | 0.0652 | 0.2553 | 0.1893 | 1.6235 |
| | 1 h | 0.0689 | 0.2625 | 0.2006 | 1.7692 |
| LSTM-AE | 30 min | 0.0166 | 0.1288 | 0.0799 | 0.7567 |
| | 1 h | 0.0218 | 0.1476 | 0.0762 | 0.7992 |
| CNN-LSTM | 30 min | 0.0142 | 0.1192 | 0.0804 | 0.6728 |
| | 1 h | 0.0182 | 0.1349 | 0.0991 | 0.7256 |
| DCNN-AM | 30 min | 0.0083 | 0.0911 | 0.0489 | 0.6118 |
| | 1 h | 0.0089 | 0.0943 | 0.0496 | 0.6412 |
| DCNN-LSTM-AE | 30 min | 0.0242 | 0.1556 | 0.0756 | 0.7128 |
| | 1 h | 0.0289 | 0.1700 | 0.0862 | 0.7196 |
| LSTM-AE-AM | 30 min | 0.0048 | 0.0693 | 0.0384 | 0.6203 |
| | 1 h | 0.0056 | 0.0748 | 0.0696 | 0.6495 |
| DCNN-LSTM-AE-AM | 30 min | 0.0041 | 0.0640 | 0.0329 | 0.5901 |
| | 1 h | 0.0081 | 0.0900 | 0.0657 | 0.6336 |

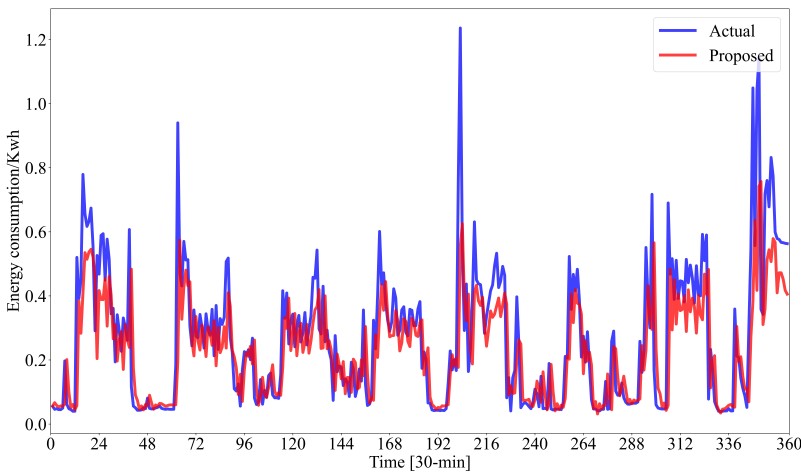

**Figure 13.** Prediction results of DCNN-LSTM-AE-AM on the SGSC dataset at a 30-min resolution.

## 5. Conclusions

This paper proposes a short-term load forecasting model to predict residential energy consumption. A hybrid electric load forecasting model, i.e., DCNN-LSTM-AE-AM, is constructed with the introduction of existing deep learning methods. We use multiple similar-days data in the vicinity of missing values as the basis for inferring the original data and use the TNN algorithm to fill in the missing data. In the initial feature-extraction stage, DCNN broadens the time horizon to retain the load features. LSTM-AE is used to improve the analysis capability of features. In the feature fusion stage, the importance of features in each period is summarized based on AM, which enhances the final prediction accuracy. The validity of the proposed method is verified on two real-world datasets. Experimental results show that the proposed method improves the accuracy of residential load forecasting and can capture low-load data features.

In future work, we need to improve the accuracy of residential load forecasting by exploiting residential lifestyle features. Moreover, how to achieve real-time forecasting through methods such as online learning is another work.

**Author Contributions:** Conceptualization, X.J.; methodology, K.Y.; software, H.H.; validation, Y.Z. and H.H.; formal analysis, X.J. and R.B.; investigation, X.J.; resources, R.B.; data curation, D.C.; writing—original draft preparation, X.J.; writing—review and editing, Z.C.; visualization, Y.Z.; supervision, Z.C. and R.B.; project administration, D.C.; funding acquisition, Z.C. All authors have read and agreed to the published version of the manuscript.

**Funding:** This research was funded by the National Natural Science Foundation of China (NSFC) under grant nos. 51874205; Jiangsu Qinglan Project.

**Institutional Review Board Statement:** Not applicable.

**Informed Consent Statement:** Not applicable.

**Data Availability Statement:** The figures and tables used to support the findings of this study are included in the article.

**Conflicts of Interest:** The authors declare no conflict of interest.

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
