# Peer review of "A Hybrid Residential Short-Term Load Forecasting Method Using Attention Mechanism and Deep Learning"

_buildings, doi:10.3390/buildings13010072_

Round 1
Reviewer 1 Report
Overall, the paper is interesting and well written.
There are some improvements to be made for final pubblication
First and foremost, the paper doesn't cite the literature that states that deep learning is often not competitive with standard ML methods
(for example, see the introduction of https://arxiv.org/abs/1908.07442).
Practically, this means that at least an ml model (such as XGBoost) should be added as benchmark model, in order to put in perspective the results.
Secondly, it should be checked and specified that there is no leakage between train and test data (some components of the model takes into account future data in training, such as biLSTM, TNN)
Also, info about training and test split should be added.
Finally, I would also complete the abduction study, in order to get what component is more important for the improvements (in practice: add LSTM-AE-AM, DCNN-LSTM-AE, DCNN-AM. Difference from proposed method would give an indication to importance of (respectively) DCNN, AM, LSTM-AE). (This applies if the training procedure doesn't take months, of course!)
Other minor corrections:
some expressions need clarification: "improve" r3, "enriched" r18, "guarantees the people life" r31, "extremely sensitive" r121, "meanings" in 208, "as possible" in 229
r32: blank space to be added: before Therefore
r34: time series forecasting problem
r41: the MOST common used methods
Figure 2 (or a similar one) should be anticipated in the introduction. It is difficult to follow the contributions without a figure at this stage of the article.
r71: existing approaches for feature extraction?
r89-90: the problem of prediction effect reduction in future climate change is addressed by deeplearning and/or your proposed model? (if so, it is not tested in the experiment result datasets)
r116: missing reference for resident load dataset
r120: "proves the effectiveness of short-term load forecasting" what do you mean? of image processing with cnn for time series forecasting?
r150: which project? Is a reference missing?
r152: specify the type of forecasting problem (which horizon? 15 and 1 hour ahead?)
r165: 1-dimension data means time-series? specify
in eq (3): w is the filter? specify
also, a standard convolution is a specific case of dilated convolution (such as with r=1?). And dilated convolution layer , is like a fully connected layer with only specific nonnull weights?
Maybe in figure 3 the standard convolution (and possibly a fully connected layer) should be given as reference for the reader to understand better
r181: how do you make sure the info taken from the future is not used at inference time?
r196: is paying -> pays more attention; is represents -> represents
r207: reference [44] doesn't work. Maybe a misspelling?
r217: are you using TNN also in the test set? if so it shouldn't. Maybe add an additional figure with the differences between training procedure and inference/test procedure?
Also, which is the train/test split?
Also add, or at least give an indication, to training time, memory of final model, inference time in 4.1.2
Figure 7: add grid (like the grid on of matlab) and align the time series to the figure box (no blank space at left and right), and add date in x-axis. Same for figures 10,11,12
Also: measures are in load/power [kW] or energy [kWh]? If kWh, it is not load. check it out.
Table 2: criteria for selection?
r231: strictly speaking, accuracy is not appropriate for any regression problem, since it is applicable only to classification problems. I get that you use the word in a broader sense, but rephrase, please.
r232-233: isn't the contrary? \psi is atual and \psi_{prev} predicted? I guess prev is pred? or is it "previous"?
r240: "with of"-->for?, "Form"-->From
r245: length of time series data refers to the chunck to be used as input? specify
Table 4: rememeber to add (at least) xgboost.
Author Response
Thanks for your valuable comments and kind suggestions. We are uploading (a) our point-by-point response to your comments (below) (response to reviewers), (b) an updated manuscript with yellow highlighting indicating changes (Supplementary Material for Review), and (c) a clean updated manuscript without highlights (Main Manuscript).

Reviewer 2 Report
The study proposed a short-term load forecasting approach integrating various machine-learning methods which outperforms conventional machine-learning/deep-learning methods. The authors describe the method clearly and present the results very well. Before accepting the paper, I have a few comments:
1. Introduction: The logic of the section is clear to readers. Only one piece of inaccuracy I would like to point out is that the authors concluded that "Support vector machine (SVM)[4], artificial neural network (ANN) [5], and long short-term memory network (LSTM)[6] are the common used methods." How did the authors get the conclusion? Is it from a reference or extracted from the authors' knowledge and experience?
2. Section 4.1.1: What's the time granularity of the SGSC? Please indicate it in this section instead of Section 4.5.
3. Figure 7: What does the value shown in the figure represent? Is it the electricity load from one house or the total of the dataset? Explain the xlabel (Times [15-min]/ Times [30-min]). Does it mean that the authors averaged the 1-minute data during 15min/30 min and plotted the mean value in the figure?
4. Section 4.1.2: It's good that the authors provided details about the hyperparameters, which are always ignored in studies' reports. Could the authors provide more information on how they determined the hyperparameters' values?
5. For other compared methods, how did the authors set the hyperparameters?

Author Response

(The authors gave the same response as above.)

Round 2
Reviewer 1 Report
the issues have been addressed. Proofread again for mistakes (e.g. r91, past particple missing: data-driven load forecasting 90 technologies have received extensively).
Author Response
Thanks for your carefully reading and kindly reminding. In the revised manuscript, we have corrected them.
1. We supplement and modify the missing or problematic expressions in the article.
2. We think we have solved all the grammar and tense problems.
3. We re-judge the addition of the definite article. Due to the large number, we only marked some key changes.